# Preparation and Characterization of Cardanol-Based Flame Retardant for Enhancing the Flame Retardancy of Epoxy Adhesives

**DOI:** 10.3390/polym14235205

**Published:** 2022-11-29

**Authors:** Won-Ji Lee, Sang-Ho Cha, Do-Hyun Kim

**Affiliations:** 1Department of Chemical Engineering, Kyonggi University, 154-42, Gwanggyosan-ro, Yeongtong-gu, Suwon-si 16227, Republic of Korea; 2Department of Fire Safety Research, Korea Institute of Civil Engineering and Building Technology, 64, 182Beon-Gil, Mado-ro, Mado-Myeon, Hwaeong-si 18544, Republic of Korea

**Keywords:** cardanol, adhesives, flame retardant, epoxides, lap shear strength

## Abstract

Epoxy resin has a versatile set of applications due to its excellent properties. However, its easily flammable property limits further applications. A bio-based flame retardant, cardanyl diphenylphosphate (CDPP), was successfully synthesized via condensation reaction between cardanol and diphenyl phosphoryl chloride. The chemical structure of CDPP was confirmed via ^1^H nuclear magnetic resonance and Fourier transform infrared spectroscopy. To overcome the flammable property of epoxy resin, different amounts of CDPP were incorporated into the epoxy resin. The thermal stability of epoxy resin with CDPP was reduced due to its phosphorus component, which had a relatively weak bond. Meanwhile, the measured char residue of epoxy resin with CDPP was increased compared to its calculated value, which indicated that CDPP promoted the formation of char residue. The limiting oxygen index of epoxy resin with CDPP was enhanced as the amount of CDPP increased from 22.1% for EP0 to 32.7% for EP10. The maximum value of the heat release rate per unit area and total heat release values of EP10 decreased by 23.23% and 12.02%, respectively, as compared to those of EP0. Additionally, single lap shear strength confirmed the improvement in the adhesion property of EP5. The lap shear strength increased to 7.19 MPa for EP5 compared to 6.27 MPa for EP0. This behavior might be due to the higher polarity of the phosphorus components. Based on the findings gathered in the present study, the incorporation of a bio-based flame retardant (CDPP) in epoxy resin has the potential for improving flame retardancy and adhesion property, which will be promising for the industrial area.

## 1. Introduction

Epoxy resin, a major thermosetting polymer possessing outstanding mechanical properties, chemical resistance, and electrical insulation properties, has versatile applications, including as coating, insulation material, construction material, and paint [1,2,3,4,5]. Epoxy adhesive has been widely employed in various applications, from automobile to building materials, owing to its excellent adhesion property [6]. However, epoxy adhesives are vulnerable to flame, thereby restricting their application in the industrial sector [7,8,9]. Recently, the flame retardancy standards for construction materials, including adhesives, have been strengthened. A flame retardant can be incorporated to epoxy adhesives to improve its capability to resist flame [10,11,12,13,14]. The use of brominated epoxy resins has been reported to be an effective strategy to endow epoxy resins with flame retardancy [15,16,17,18]. Halogen-based flame retardants are inexpensive and exhibit excellent flame retardancy even when added in small amounts. However, halogenated flame retardant is known to release toxic and corrosive smoke harmful to humans [19,20,21,22]. For instance, brominated flame retardants release highly toxic compounds, such as dibenzofurans and dibenzodioxins, which have deleterious effects on human health [23]. The addition of an additive-type flame retardant, such as phosphorus-based, nitrogen-based, and boron-based flame retardants, is a facile method for improving the flame retardancy of epoxy adhesive without incorporating halogenated flame retardants [24,25,26,27]. Phosphorus-containing flame retardants impart excellent flame retardancy to epoxy resin and do not produce toxic compounds during combustion. Recently, the interest in eco-friendly methods which preserve human health has fueled the demand for flame retardants derived from renewable resources [28,29,30,31]. Current research has focused on the development of highly efficient and less toxic flame retardants derived from bio-based compounds [32]. Bio-based flame retardants use various biomasses, from the simple hydroxyl group containing compounds such as cardanol, eugenol, and vanillin, to compounds containing long alkyl chains, such as soy bean oil [33,34,35]. Cardanol, obtained from cashew nut shell liquid, has been researched as an inexpensive and readily available bio-based source. Moreover, cardanol is considered an interesting chemical candidate for the production of versatile value-added materials due to its unique structure with an unsaturated double bond and hydroxyl group [36]. In a previous study, some bio-based materials were used as green adhesives [37,38]. However, relatively few studies have reported on bio-based additives that simultaneously improve flame retardancy and adhesion property. In this study, cardanyl diphenylphosphate, CDPP, an eco-friendly additive-type flame retardant used in epoxy adhesive composites, was successfully synthesized using cardanol. The addition of CDPP improved the flame retardancy and adhesive property of the epoxy adhesive composites.

## 2. Materials and Methods

### 2.1. Materials

Cardanol was supplied by Biochempia (Korea) and was used as received. Diphenyl phosphoryl chloride (DPPC), triethylamine (TEA), bisphenol A diglycidyl ether (BADGE), and 4,4′-diaminodiphenylmethane (DDM) were purchased from Sigma-Aldrich. Dimethylformamide (DMF), methylene dichloride (MC), and magnesium sulfate were purchased from Daejung chemical. All the other reagents and solvents were used without further purification.

### 2.2. Synthesis of Cardanyl Diphenylphosphate (CDPP)

Phosphorus flame retardant containing bio-based cardanol was synthesized as follows. Cardanol 20 g (0.07 mol), TEA as a catalyst 8.14 g (0.08 mol), and MC 200 mL were added to a round-bottom flask. Diphenyl phosphoryl chloride 18 g (0.07 mol), dissolved in MC 200 mL, was slowly added to the solution using a dropping funnel. The reaction was carried out at room temperature for 24 h. Subsequently, the formed TEA salt was removed using a filter paper and the trace amount of TEA salt was washed with deionized water. The moisture of the organic layer was dried with anhydrous magnesium sulfate, and then dried using a rotary evaporator. The product obtained as brownish liquid was referred to as cardanyl diphenylphosphate (CDPP).

### 2.3. Preparation of Epoxy Adhesive Series with CDPP

The preparation of the epoxy adhesive series containing CDPP was carried out as follows. Firstly, epoxy resin DGEBA and curing agent DDM were dissolved in DMF (5 mL). As shown in Table 1, the equivalent ratio of DGEBA and DDM was fixed to 1. The two solutions were mixed uniformly. Subsequently, bio-based flame retardant CDPP (0, 5, and 10 wt%) was added to the respective homogeneous solutions. The solution was poured into a silicon mold, and was cured at 100 °C for 2 h and 140 °C for 4 h. The cured product was referred to as EP0, EP5, and EP10 depending on the amount of CDPP. The composition of epoxy adhesive series is summarized in Table 1.

### 2.4. Characterization

The chemical structure of bio-based flame retardant CDPP was characterized using ^1^H nuclear magnetic resonance (^1^H NMR) and Fourier transform infrared (FT-IR) spectroscopy. ^1^H NMR analysis was performed using a Bruker AVANCE-400 Hz with DMSO-d_6_ as the solvent. The FT-IR spectra were recorded at 400–4000 cm^−1^ using a Bruker Alpha-Platinum attenuated total reflectance (ATR)-FTIR spectrometer. The thermal properties were measured with thermogravimetric analysis (TGA) using a Perkin Elmer TGA 4000 instrument. The TGA measurement was carried out from 50 to 700 °C at a heating rate of 10 °C/min under a nitrogen atmosphere. Scanning electron microscopy (SEM) images of residual char were observed from JEOL JSM-7601F PLUS (accelerating voltage: 15.0 kV, working depth: 8.0 mm). The UL94 vertical burning test and limiting oxygen index (LOI) were conducted to confirm the flame retardancy of epoxy adhesive series with CDPP. The LOI of the samples was measured using a FT-LOI-404 (FESTEC) test according to the ASTM D2863 standard. The sample size was 100 × 10 × 3 mm^3^. The combustion properties of EP series were assessed using a cone calorimeter (FESTEC(FR-CC-105)) as per the procedure described by ISO 5660-1 standard. The sample, 100 × 100 mm^2^ plates measuring 7.0–7.5 mm thick wrapped in aluminum foil, was horizontally exposed to an external heat flux of 50 kWm^−2^ for 600 s. Various parameters were measured including the maximum value of the heat release rate per unit area (Max-HRR) and total heat release (THR) during 0–600 s. The experimental error of the cone calorimeter data was about 5%. Single lap shear tests were carried out to confirm the effect of CDPP on the adhesion property of epoxy adhesive series. The epoxy adhesives were applied to wood plate (40 × 10 × 3 mm^3^) with a bonding area of (5 × 10 mm^2^), and were then hot-pressed at 160 °C, 2 MPa for 5 min. Shear strength measurements were performed using the universal testing machine (QM100S) at the tensile speed of 5 mm/min. The reported results are the average value of at least four replicates.

## 3. Results

### 3.1. Synthesis and Characterization of CDPP

Bio-based phosphorus flame retardant which could be applied to epoxy adhesive was synthesized by the reaction between cardanol and diphenyl phosphoryl chloride using TEA as a catalyst (Figure 1). The chemical structure of CDPP was confirmed using ^1^H NMR and FT-IR.

The peaks at 7.05 and 6.54 ppm, corresponding to the protons of the cardanol phenyl group, were shifted to the down-field area to 7.34 and 7.11 ppm in CDPP, demonstrating the occurrence of the condensation reaction between DPPC and cardanol. In addition, the characteristic proton peaks of cardanol, attributed to unsaturated double bonds at 5.31–5.38 ppm and attributed to alkyl chains at 3.35–0.85 ppm, were also observed in CDPP (Figure 2).

To further confirm the chemical structure of CDPP, FT-IR measurements were carried out as shown in Figure 3. The characteristic peaks of cardanol corresponding to C=C bonds on benzene ring (1594, 1455 cm^−1^), hydrogen atoms of aromatic ring (775, 694 cm^−1^), and unsaturated hydrocarbon (3010 cm^−1^) were observed in CDPP [39,40]. The peak at 958–1030 cm^−1^ could be ascribed to P-O-Ph. The peak at 1173 cm^−1^, corresponding to the stretching vibration of P=O, was shifted from 1173 to 1185 cm^−1^ [41]. In addition, the peaks corresponding to the phenolic hydroxyl group (3100–3500 cm^−1^) of cardanol and P-O-Cl (542–587 cm^−1^) of diphenyl phosphoryl chloride were absent in CDPP [42]. Therefore, we believe that CDPP was successfully synthesized from the reaction between DPPC and cardanol.

### 3.2. Thermogravimetric Curves of CDPP and EP Series

The thermal properties of CDPP and EP series containing CDPP were evaluated by thermogravimetric analyses (TGA). As shown in Figure 4, CDPP was degraded via a two-step process. The first thermal decomposition stage was initiated at 220 °C and the T*_d_*_30%_ (temperature at 30 wt% weight loss) was observed at 344 °C, which might be attributed to the decomposition of the phosphorus group and alkyl chain in CDPP. Flame retardant compounds with a phosphorus group generally initiate thermal degradation around 300–350 °C [43,44]. Subsequently, the second thermal decomposition stage of CDPP was due to the degradation of benzene ring in cardanol.

Additionally, the effect of CDPP on the thermal properties of the EP series was evaluated using TGA. Several characteristic temperatures, such as T*_d_*_5%_ (temperature at 5 wt% weight loss), T*_d_*_30%_ (temperature at 30 wt% weight loss), and char residue at 700 °C are summarized in Table 2. It was found that the thermal stability of EP series decreased with the introduction of CDPP. For example, T*_d_*_30%_ of EP10 was 379 °C, compared to 394 °C for EP0. This result is attributed to the phosphorus compound of CDPP, which has a relatively lower degradation temperature than epoxy matrix.

CDPP, containing a phosphate group, was decomposed to form phosphoric acid compound with strong dehydration effect, thereby catalyzing the dehydration of the EP series into the char layer. A significant amount of char residue was observed in the EP series at high temperature. It was confirmed that the amount of char residue increased as the content of CDPP increased. For instance, the measured char residue value of EP10 was 23.72% as compared to 17.69% for EP0. The effect of CDPP on the formation of char layer could be further supported by the calculated char residue [45]: Calculated char residue of EP series containing CDPP = (2.41 × flame retardant content) + (17.69 × EP0 content)(1)

In this equation, 2.41 and 17.69 refer to the measured char residue of CDPP and EP0, respectively, at 700 °C in TGA. In EP series, each measured char residue showed a higher value than the corresponding calculated value. For instance, while the calculated char residue of EP10 was 16%, the measured char residue was 24%. Therefore, it was suggested that the introduction of CDPP could have induced the formation of a relatively larger amount of char from the carbonization of EP series [46]. Considering that the degree of char formation is a key factor determining the efficiency of the flame retardants, it was expected that CDPP could increase the flame retardancy of EP series.

### 3.3. Flame Retardancy of EP Series with CDPP

The UL94 vertical burning test was carried out to clarify the flame retardancy of the EP series. A vigorous and fierce flame, as well as a large amount of drips, was observed within a short time for the EP0 sample. EP5 and EP10 achieved a V-1 rating in the UL94 vertical burning test, which suggests that CDPP enhanced the flame retardancy of EP series. As shown in Figure 5, LOI was used to confirm of the effect of CDPP on the flame retardancy of the EP series. Generally, polymer materials with a LOI value higher than 26% can be regarded as significant flame retardancy materials [47]. It was observed that the LOI value of the EP series significantly increased (from 22.1% of EP0 to 32.7% of EP10) after incorporating CDPP. This could be due to the unique structure of CDPP, which possesses a self-crosslinkable long alkyl chain and phosphate group.

It could be inferred that self-crosslinked CDPP imparted an enhanced flame retardancy to the EP series. Peng et al. reported that self-crosslinked flame retardant endowed a better flame retardancy to the EP matrix. The self-crosslinked flame retardant could promote the carbonization process of the EP matrix and the formation of a tough char layer, which could retard fire diffusion [48]. Likewise, cardanol, which is one of the components of CDPP, is known to be self-crosslinked under thermal treatment [49]. This self-crosslinking character of CDPP can be confirmed by FTIR. The unsaturated bond peak of CDPP originating from cardanol at 3010 cm^−1^ disappeared after epoxy curing, as shown in Appendix A.

Additionally, the cone calorimeter test, which is one of the general tools for flame retardancy evaluation, was conducted. The cone calorimeter was measured based on the correlation between the amount of heat generated during combustion and the amount of oxygen consumed [50]. Table 3 depicts the maximum value of the heat release rate (Max-HRR) with respect to time (s), total heat release (THR), and weight loss ratio, which are important factors when evaluating the flame retardancy property. Figure 6 shows the HRR curves. The Max-HRR value of the EP series declined with increasing amount of CDPP. The Max-HRR value of EP0 was 564.07 kW/m^2^, indicating fierce combustion. In contrast to EP0, the Max-HRR value of EP10 was 433.03 kW/m^2^. In other words, as the amount of CDPP decreased, the time (s) at which the Max-HRR appeared was delayed. When the amount of CDPP increased (EP5, EP10), the Max-HRR was measured within 85 s, and low HRR was measured until the end of the test. The HRR value over time and the measured Max-HRR and time in seconds can predict the fire hazard potential of a material. Therefore, it can be argued that the higher the value of Max-HRR and the slower the measurement time, the higher the fire risk with regard to continuous combustibility. In the opposite case, it was observed that the fire risk was relatively low, and this finding was consistent with the TGA char residue results. It was found that CDPP conferred an enhanced flame retardancy to the EP series.

To confirm the effect of the phosphorus group in CDPP on the formation of residual char, the morphology of residual char was observed using SEM after the combustion of epoxy resin. Figure 7a–c show the result of the cone calorimeter for the EP series. The char residue of EP0 exhibited a fractured structure with many voids and holes, while EP5 and EP10 exhibited relatively compact surfaces. The morphology of char residue was further confirmed by SEM (Figure 7d–f). The char layer of EP0 presented many cavities, which allowed the transfer of flammable materials. For EP5, the cavities were clogged by a dense char layer. EP10 exhibited a stable char layer after combustion, which obstructed the transfer of heat and volatile.

In general, the phosphorus-based flame retardant conferred an enhanced flame retardancy to materials due to the combination of the gas-phase mechanism and condensed-phase mechanism. In the gas-phase mechanism, the flame retardants capture free radicals, such as hydroxyl and hydrogen radicals, which are needed for combustion. In the condensed mechanism, phosphorus-based flame retardants promote the generation of phosphoric acid and the formation of the char layer, which inhibits heat transfer to the polymer matrix. In the EP series containing CDPP, the measured char residue value was significantly higher than the calculated char residue. In addition, the Max-HRR, THR value, and weight loss ratio decreased as the amount of CDPP increased. These results suggest that self-crosslinked CDPP promoted the carbonization of EP series and the formation of the char layer, which inhibited the flame [51,52,53,54].

### 3.4. Adhesion Property of EP Series Containing CDPP

The adhesion property of the EP series was assessed by a lap shear strength test. The EP series containing CDPP were applied to a wood plate with a specific bonding area (5 × 10 mm^2^). The dimensions for the single lap shear test of EP series was shown in Figure 8.

The EP series was thermally cured by hot pressing at 160 °C, 2 MPa for 5 min. Lap shear strength was calculated using the following equation: Lap shear strength = F_max_/A(2)
where F_max_ and A correspond to the maximum shear force and overlapping adhesive area, respectively [55]. It was observed that the lap shear strength value increased from 6.27 MPa of EP0 to 7.18 MPa of EP5 (Table 4) when CDPP was used. This improvement was related to the addition of CDPP, which has a high polarity due to its phosphorus components. The higher polarity of the adhesive component causes the greater adhesion property [56,57]. However, a reduction of lap shear strength was observed from 7.18 MPa of EP5 to 5.55 MPa of EP10. According to the lap shear strength results, the adhesion science needs for consideration to multi-disciplined topic, which incorporate surface chemistry, physics, rheology, polymer chemistry, and other aspects [58]. It might be inferred that other factors, such as mechanical interlocking, were more influential on the adhesion property of EP10 in comparison to polarity. The excessive addition of self-crosslinked CDPP caused the reduction of the mechanical interlocking as well as the inhibition of hydrogen bonding between the wood substrate and the epoxy matrix, leading to the reduced adhesion property of EP10 [59].

## 4. Conclusions

CDPP, a renewable-resource-based flame retardant, was successfully synthesized and applied to epoxy adhesive series. The thermal stability of the EP series containing CDPP decreased due to the earlier decomposition of phosphorus compounds which possessed relatively weak bonds. On the other hand, the measured char residue value was higher than the calculated char residue value. This was due to the self-crosslinkable CDPP, which promoted the formation of the char layer of the EP series. Self-crosslinking CDPP was confirmed via FT-IR measurement. The flame retardancy of the EP series was improved by increasing CDPP content. EP5, containing a small amount of CDPP, exhibited a significant LOI value of 28.2%. Moreover, the cone calorimeter test showed a similar fire-retardant effect. The Max-HRR value was decreased from 564.07 kW/m^2^ at 348 s of EP0 to 433.03 kW/m^2^ at 83 s of EP10. Additionally, the THR value was decreased from 168.1 MJ/m^2^ of EP0 to 147.9 MJ/m^2^ of EP10. The improvement of flame retardancy was caused by the compact char layer, which acted as an effective barrier. It was confirmed that the addition of CDPP to the EP series improved the adhesion property. The lap shear strength value was improved from 6.27 MPa of EP0 to 7.18 MPa of EP5, due to the polarity of the phosphorus components. It was considered that EP adhesive series containing CDPP exhibited flame retardancy and improved adhesion properties, supporting its use for various applications.

## Figures and Tables

**Figure 1 polymers-14-05205-f001:**
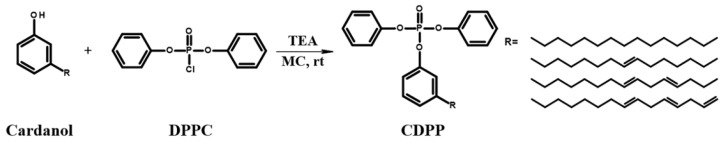
Synthesis route of CDPP.

**Figure 2 polymers-14-05205-f002:**
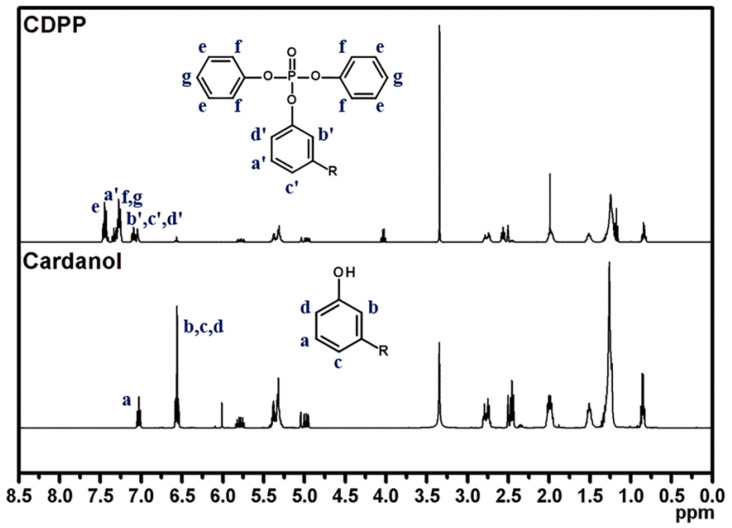
^1^H NMR spectra of cardanol and CDPP.

**Figure 3 polymers-14-05205-f003:**
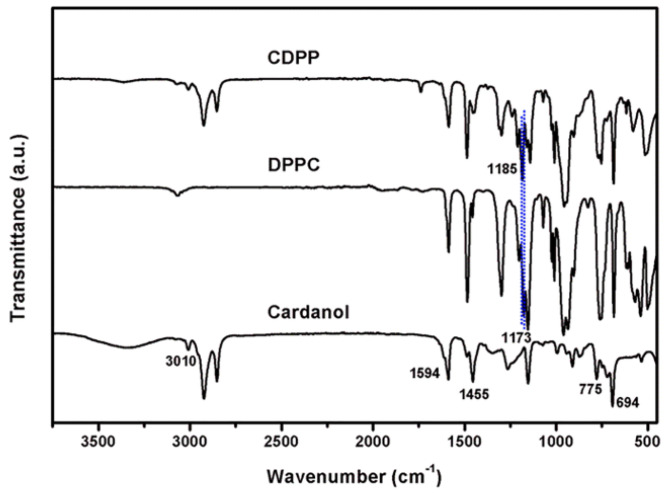
FT-IR spectra of cardanol, DPPC, and CDPP.

**Figure 4 polymers-14-05205-f004:**
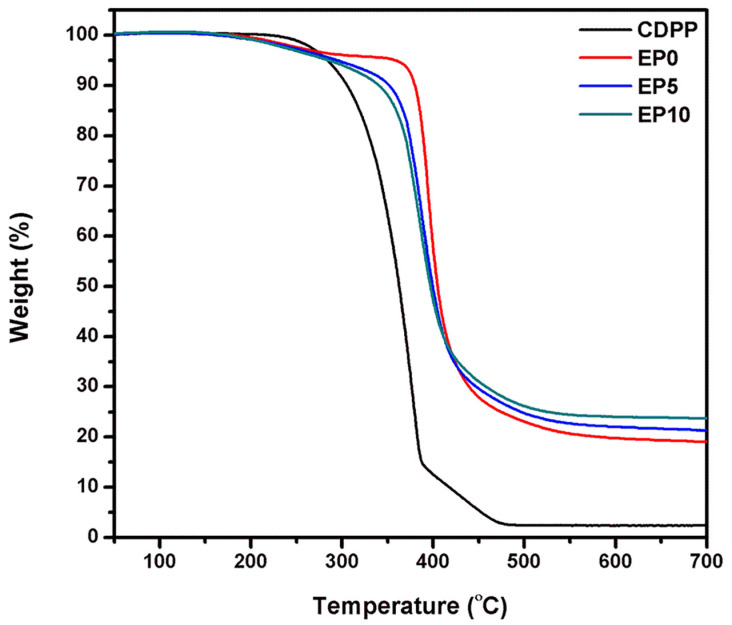
TGA curves of CDPP and EP series in nitrogen atmosphere.

**Figure 5 polymers-14-05205-f005:**
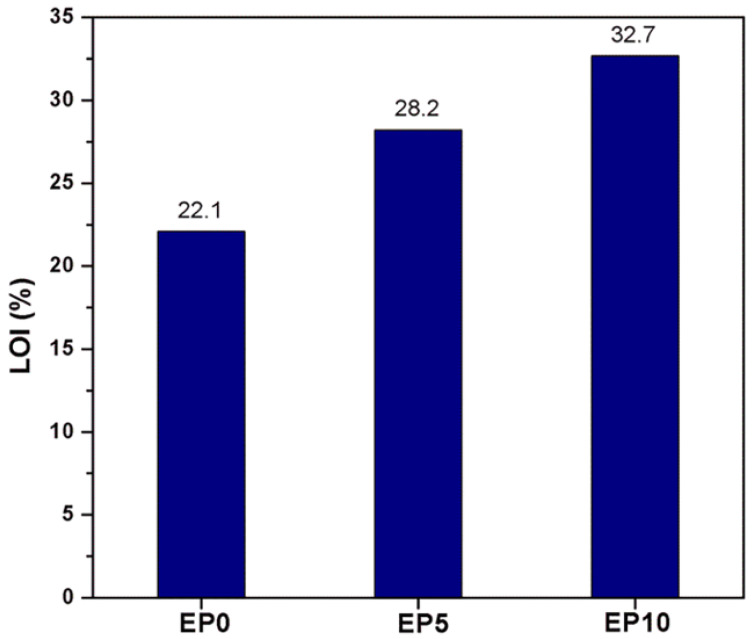
LOI results of EP series.

**Figure 6 polymers-14-05205-f006:**
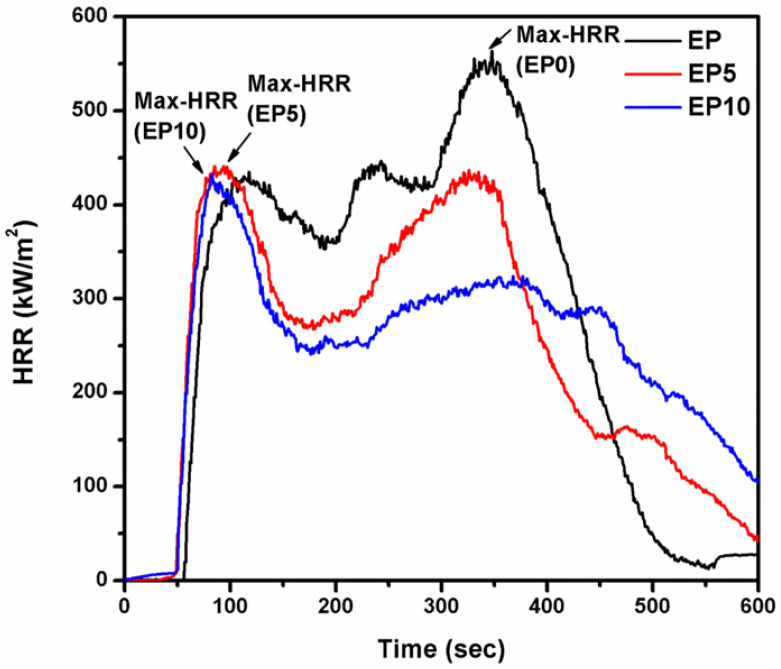
Max-HRR points versus burning time for EP series.

**Figure 7 polymers-14-05205-f007:**
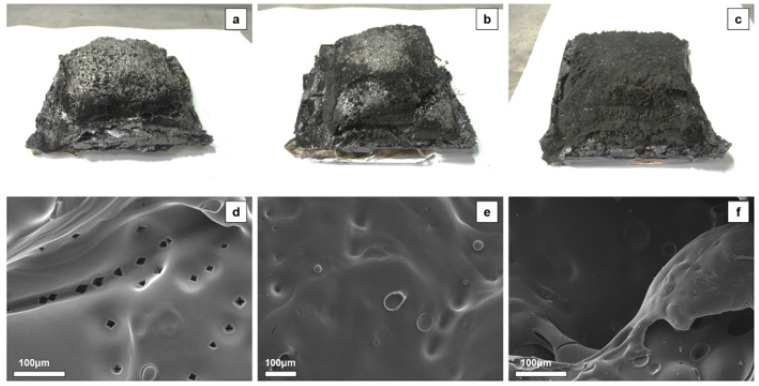
Digital photos of residual char for (**a**) EP0, (**b**) EP5, (**c**) EP10 and SEM images of residual char for (**d**) EP0, (**e**) EP5, and (**f**) EP10 after cone calorimeter test.

**Figure 8 polymers-14-05205-f008:**
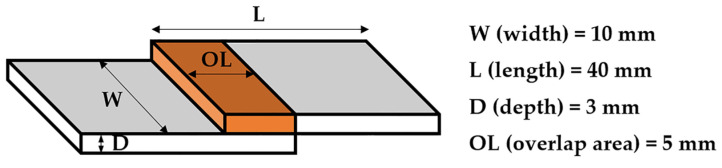
Dimensions for single lap shear test of EP series.

**Table 1 polymers-14-05205-t001:** Formulation of the epoxy adhesives with different CDPP contents.

Samples	BADGE (wt%)	DDM (wt%)	CDPP (wt%)
EP0	77.64	22.36	0
EP5	73.75	21.24	5.01
EP10	69.83	20.11	10.06

**Table 2 polymers-14-05205-t002:** Thermogravimetric data of CDPP and EP series.

Samples	T*_d_*_5%_ (°C)	T*_d_*_30%_ (°C)	Residue % at 700 °C	
			Measured	Calculated
CDPP	284	344	2	-
EP0	359	394	18	-
EP5	295	383	21	17
EP10	285	379	24	16

**Table 3 polymers-14-05205-t003:** Cone calorimeter data of EP series.

Samples	Max-HRR (kW/m^2^) at Time (s)	THR (MJ/m^2^)
EP0	564.07 (348)	168.1
EP5	441.34 (85)	148.1
EP10	433.03 (83)	147.9

**Table 4 polymers-14-05205-t004:** Single lap shear test results for EP series.

Samples	Lap Shear Strength (MPa)
EP0	6.27 ± 0.18
EP5	7.18 ± 0.35
EP10	5.55 ± 0.41

## Data Availability

The data presented in this study are available on request from the corresponding author.

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
