# Peer review of "Preparation and Characterization of Cardanol-Based Flame Retardant for Enhancing the Flame Retardancy of Epoxy Adhesives"

_polymers, 2022, doi:10.3390/polym14235205_

Round 1

Reviewer 1 Report

In this work, the authors reported a bio-based, P-containing flame retardant for epoxy resin. The reported epoxy resins showed improved flame retardancy and adhesion properties. This work is well-organized, but the uploaded manuscript needs major revision before publication. The specific issues are listed as follows.

1.     When the abbreviation first appears, its full name should be provided, such as ‘EP0’, ‘EP10’, ‘pk-HRR’ and ‘THR’ in abstract.

2.     In introduction, ‘Halogen-based flame retardants … and corrosive smoke harmful to humans’, some references should be cited to support this point, such as Chem. Eng. J., 2023, 451: 138768; Polymer, 2022, 261: 125399; React.  Funct. Polym., 2022, 178: 105335; & Compos. Part B: Eng., 2022, 234: 109701.

3.     In introduction, ‘Phosphorus-containing flame retardants … fueled the demand for flame retardants derived from renewable resources’, some works on bio-based P-containing flame retardants should be cited, such as Carbohy. Polym, 2022, 298: 120141; ACS Sustainable Chem  Eng, 2022, 10(15): 5055-5066; Polym Adv Technol, 2022, 33(3): 770-781; & Compos Part A, 2022, 160: 107028.

4.     The introduction should be revised, which should point out the flaws in previous works and the novelty of this work.

5.     In table 1, the unit of BADGE, DDM and CDPP should be changed into ‘wt%’.

6.     The English of this manuscript should be polished.

7.     In the first paragraph of section 3.3, the full name of ‘Td30%’ is missed, and the first decomposition stage of CDPP might be not only due to the decomposition of phosphorus group. The degradation of alkyl chain segment might also occur in this stage.

8.     In Table 2, displays the contents rounded to one decimal place.

9.     In section 3.3, ‘This could be due to the unique structure of CDPP which possess self-cross linkable long alkyl chain and phosphate group’, such description is incorrect, the alkyl chain is usually flammable.

10.  In addition to heat release, how about the smoke generation of the reported EPs?

11.  The UL-94 ratings of the reported EP systems should be provided.

12.  In the section 3.3, ‘In general, the phosphorus-based flame retardant conferred … which inhibited the flame’, some references should be cited to support this point, such as ACS Appl. Polym. Mater., 2022, 4(5): 3564-3574; Chem. Eng. J., 2019, 375: 121916;  SusMat, 2022, 2(4): 411-434; Chem. Eng. J. 2022, 430, 132712.

13.  Please double-check the reference format.

Author Response

Dear Editor

We are submitting a revised manuscript entitled “Preparation and characterization of cardanol-based flame retardant for enhancing the flame retardancy of epoxy adhesives”.

We thoroughly read your mail and reviewer’s comments. The kind and meticulous comments of the reviewers were very helpful to our manuscript. In addition, thanks to reviewers sincerely comments, the missing part could be confirmed. Therefore, we included the reviewer’s comments in the manuscript as follows (The revised or added parts in manuscript were written with red letters).

Point 1: When the abbreviation first appears, its full name should be provided, such as ‘EP0’, ‘EP10’, ‘pk-HRR’ and ‘THR’ in abstract.

Response 1: As the reviewer’s kindly commented, we added the full name for ‘pk-HRR’ and ‘THR’. Unfortunately, due to the abstract section character limit, the full name of EP0, EP5, and EP10 could not be added.

Point 2: In introduction, ‘Halogen-based flame retardants … and corrosive smoke harmful to humans’, some references should be cited to support this point, such as Chem. Eng. J., 2023, 451: 138768; Polymer, 2022, 261: 125399; React.  Funct. Polym., 2022, 178: 105335; & Compos. Part B: Eng., 2022, 234: 109701.

Response 2: Thanks to reviewer’s meticulous comments, some references, mentioned by reviewer were added to support the corresponding sentence.

Point 3: In introduction, ‘Phosphorus-containing flame retardants … fueled the demand for flame retardants derived from renewable resources’, some works on bio-based P-containing flame retardants should be cited, such as Carbohy. Polym, 2022, 298: 120141; ACS Sustainable Chem  Eng, 2022, 10(15): 5055-5066; Polym Adv Technol, 2022, 33(3): 770-781; & Compos Part A, 2022, 160: 107028.

Response 3: Thank you for reviewer’s kind comments, it was very helpful to make up for this paper. All references attached are added to the manuscript.

Point 4: The introduction should be revised, which should point out the flaws in previous works and the novelty of this work.

Response 4: According to reviewer’s kindly comments, the flaws in previous works and the novelty of this work was added in the introduction section, which emphasized the importance of this research more than before. The added sentences were marked in red in the draft.

; In the previous study, some bio-based materials were used for green adhesive. However, relatively few studies have been reported on bio-based additives that simultaneously improve flame retardancy and adhesion property. 

Point 5: In table 1, the unit of BADGE, DDM and CDPP should be changed into ‘wt%’.

Response 5: Thanks to reviewer’s carefully comments, in table 1, the unit of components was changed to ‘wt%’.

Point 6: The English of this manuscript should be polished.

Response 6: Thanks to reviewer’s kind comments, the English of this manuscript was checked again.

Point 7: In the first paragraph of section 3.3, the full name of ‘Td30%’ is missed, and the first decomposition stage of CDPP might be not only due to the decomposition of phosphorus group. The degradation of alkyl chain segment might also occur in this stage.

Response 7: Thanks to reviewer’s meticulous comments, the full name of ‘Td30%’ was added in manuscript.

We wanted to real-time FTIR measurement to confirm the thermal decomposition stage of CDPP. However, it was difficult because its state is liquid. Therefore, the thermal decomposition stage of CDPP was inferred using the real-time FTIR of EP10 (EP adhesive containing 10 wt% CDPP).

Figure. Real-time FT-IR of EP10

First, the peaks (1024-1009 cm-1) attributed to phosphorus group of CDPP were weakened. Subsequently, it was observed that the peak (1608 and 1504 cm-1) corresponding to the benzene ring of cardanol was weakened together at 300 °C. Based on this result, it is described in the text that the decomposition of phosphorus group and alkyl chain proceeds first, followed by the decomposition of benzene ring of cardanol.

Point 8: In table 2, displays the contents rounded to one decimal place.

Response 8: Thanks to reviewer’s carful comments, the contents in table 2 was changed.

Point 9: In section 3.3, ‘This could be due to the unique structure of CDPP which possess self-cross linkable long alkyl chain and phosphate group’, such description is incorrect, the alkyl chain is usually flammable.

Response 9: Generally, alkyl chain is relatively flammable than other chemical compound containing ring structure (for instance, benzene ring and furan ring). However, in this paper, the unsaturated bond of cardanol was self-crosslinked during epoxy curing process, which was confirmed via FT-IR measurement (Figure S1). The crosslinking of cardanol unsaturated bond via thermal treatment could be supported by other reference (ref; Kim, S.-H.; Kim, S.-B.; Cha, S.-H. Preparation and Characterization of Biopolyurethane Film with a Novel Cross-linkable Biopolyol Based on Cardanol. Polymer(Korea) 2018, 42, 736-746.). The cross-linked flame retardant can induce the improvement of flame retardancy (ref; Peng, X.; Li, Z.; Wang, D.; Li, Z.; Liu, C.; Wang, R.; Jiang, L.; Liu, Q.; Zheng, P. A facile crosslinking strategy endows the traditional additive flame retardant with enormous flame retardancy improvement. Chemical Engineering Journal 2021, 424, 130404.).

Point 10: In addition to heat release, how about the smoke generation of the reported EPs?

Response 10: In this study, we did not record the smoke generation of the reported EPs, because we focus on the heat release rate (Cone calorimeter method) are more important factors in terms of fire safety mechanism.

Point 11: The UL-94 ratings of the reported EP systems should be provided.

Response 11: According to reviewer’s kind comments, the UL-94 ratings of EP series were added in the draft.

; In section 3.3. The UL94 vertical burning test was carried out to clarify the flame retardancy of EP series. The vigorous and fierce flame as well as a large amount of drips was observed within a short time for EP0 sample. For EP5 and EP10, it achieved V-1 rating in the UL94 vertical burning test, which suggests that CDPP enhanced the flame retardancy of EP series.

Point 12: In the section 3.3, ‘In general, the phosphorus-based flame retardant conferred … which inhibited the flame’, some references should be cited to support this point, such as ACS Appl. Polym. Mater., 2022, 4(5): 3564-3574; Chem. Eng. J., 2019, 375: 121916;  SusMat, 2022, 2(4): 411-434; Chem. Eng. J. 2022, 430, 132712.

Response 12: Thanks to reviewer’s meticulous comments, all references were added in manuscript.

Point 13: Please double-check the reference format.

Response 13: The reference format was checked according to reviewer’s careful comments. The reference formats were written according to the MDPI style.

Reviewer 2 Report

Comments

L. 78. There are no data on the physical state and properties of the synthesized cardanyl diphenylphosphate

L. 82. No information available on solvent concentration. There is also no information on its removal or presence in the cured epoxy adhesive.

L. 110-113. Do the adhesion test parameters follow any standard?

L. 151. The article should be supplemented with a figure with DTG curves

L. 163-165. thermal stability does not change at different temperatures. What parameters of thermal destruction in TGA analysis are we talking about here?

L. 171. Term theoretical char residue should not be used as there is no theory involved. This is a calculated char residue in accordance with the principle of additivity.

L. 175. Instead of ratio, content, percentage, or some other similar term should be used.

L. 204 et seq. The discussion about the curves is not correct, since the curve clearly shows two stages of sample combustion and two groups of corresponding peaks. We should talk about each group of peaks separately.

L. 242-246. The mention of mechanisms should be reversed, since any combustion starts from the gas phase.

L. 267. Wetting cannot affect other than polarity, since wetting directly depends on the polarity of the contacted objects. At the same time, there is no information about the change in the viscosity of the epoxy adhesive when a flame retardant is added.

L. 272. Instead of figure 8 there should be a table of results. Graphs on 3 points do not build. Statistical data should also be provided.

Author Response

Response to Reviewer 2 Comments

Dear Editor

We are submitting a revised manuscript entitled “Preparation and characterization of cardanol-based flame retardant for enhancing the flame retardancy of epoxy adhesives”.

We thoroughly read your mail and reviewer’s comments. The kind and meticulous comments of the reviewers were very helpful to our manuscript. In addition, thanks to reviewers sincerely comments, the missing part could be confirmed. Therefore, we included the reviewer’s comments in the manuscript as follows (The revised or added parts in manuscript were written with red letters).

Point 1: # L. 78. There are no data on the physical state and properties of the synthesized cardanyl diphenylphosphate.

Response 1: Thanks to reviewer’s careful comments, the state of synthesized cardanyl diphsnylphosphate was added in manuscript.

; ‘The product obtained was referred to as cardanyl diphenylphosphate (CDPP).’ was changed to ‘The product obtained as brownish liquid was referred to as cardanyl diphenylphosphate (CDPP).’.

Point 2: # L. 82. No information available on solvent concentration. There is also no information on its removal or presence in the cured epoxy adhesive.

Response 2: According to reviewer’s kind comments, information on the amount of DMF solvent used was added in section 2.3. Also, the presence of DMF solvent in the cured epoxy adhesive was confirmed using FT-IR measurements.

As shown in the figure, it was confirmed that no DMF solvent remained in the cured EP0, EP5, and EP10.

Point 3: # L. 110-113. Do the adhesion test parameters follow any standard?

Response 3: Thanks to reviewer’s kind comments, the adhesion test parameters was checked again. In this paper, the single lap shear test was followed by ASTM D1002. However, it was adjusted while maintaining the ratio for lab scale experiments. For reliability of the single lap shear test, the experiments were repeated at least four times.

Point 4: #L. 151. This article should be supplemented with a figure with DTG curves.

Response 4: Following reviewer’s kind comments, DTG curves graph could be described as follows. The detailed data related to DTG curves was clarified in Table 2.

Point 5: #L. 163-165. Thermal stability does not change at different temperature. What parameter of thermal destruction in TGA analysis are we talking about here?

Response 5: Thanks to reviewer’s meticulous comments, this sentence was removed because it was judged to be confusing. Reviewers comments make this manuscript clearer than before.

Point 6: #L. 171. Term theoretical char residue should not be used as there in no theory involved. This is a calculated char residue in accordance with the principle of additivity.

Response 6: According to reviewer’s meticulous comments, all of these expressions have been changed.

; In abstract, ‘Meanwhile, the measured char residue of epoxy resin … compared to its theoretical value,’ was changed into ‘… compared to its calculated value,’

; L. 171 ‘The effect of CDPP on the formation of … by the theoretical char residue …’

was changed to ‘The effect of CDPP on the formation of … by the calculated char residue ….’.

Point 7: #L. 175. Instead of ratio, content, percentage, or some other similar term should be used.

Response 7: Thanks to reviewer’s careful comments, ‘ratio’ in line 175 was changed into ‘content’. The modified expressions were marked in red in the draft.

Point 8: #L. 204et seq. The discussion about the curves is not correct, since the curve clearly shows two stages of sample combustion and two groups of corresponding peaks. We should talk about each group of peaks separately.

Response 8: In this study, we would like shows the important fact of the maximum value of the heat release rate per unit area (Ref. ISO 5006-1:2015, Table 1). So we revise abbreviation according to your opinion, we rewrite pk-HRR to Max-HRR.

Point 9: #L. 242-246. The mention of mechanisms should be reversed, since any combustion starts from the gas phase.

Response 9: Thanks to reviewer’s careful comments, the mention of mechanisms in line 242-246 was reversed, which makes natural sentences than before. The modified sentences were marked in red in the draft.

Point 10: #L. 267. Wetting cannot affect other than polarity, since wetting directly depends on the polarity of the contacted objects. At the same time, there is no information about the change in the viscosity of the epoxy adhesive when a flame retardant is added.

Response 10: Thanks to reviewer’s meticulous comments, the results of single lap shear strength was carefully checked again. As the addition amount of CDPP increases, the mechanical interlocking between adhesive and substrate was weakened, resulting in the reduction of single lap shear strength. Therefore, ‘wetting’ expression was changed to ‘mechanical interlocking’. The modified expression was marked in red in the draft.

; It might be inferred that other factor, such as mechanical interlocking, was more dominant on the adhesion property of EP10 in comparison to polarity. The excessive addition of self-crosslinked CDPP caused the reduction of the mechanical interlocking as well as … .  

In this study, we did not confirm the change in the viscosity, because we focus on the catch the single lap shear strength tendency according to increasing the amount of bio-based flame retardant in EP series. However, above the appropriate amount, it is expected that the viscosity of the epoxy adhesive will be decreased, as increasing the addition of CDPP. 

Point 11: #L. 272. Instead of figure 8 there should be a table of results. Graphs on 3 points do not build. Statistical data should be provided.

Response 11: According to reviewer’s comments, figure 8 was changed into table 4. The standard deviation value was added using plus-minus sign. The results are the average value of at least four replicates.

; Table 4. Single lap shear test results for EP series

Samples

Lap shear strength (MPa)

EP0

6.27 ± 0.18

EP5

7.18 ± 0.35

EP10

5.55 ± 0.41

Reviewer 3 Report

The paper presents an interesting approach based on the Preparation and characterization of cardanol-based flame retardant for enhancing the flame retardancy of epoxy adhesives. However, the innovation of the current research work should be further highlighted and emphasized. At the same time, the authors should consider the following comments to greatly improve the quality of the paper.

1. In the abstract, add a final statement that highlights the importance of this research and its possible potentials. Also, introduce the problem in the initial lines of the abstract.

2. The introduction needs to be improved by relating to the mechanics of the studied materials and their mechanical characteristics. The references to be included are: 10.1177/0021998318790093, 10.1016/j.polymertesting.2017.09.009, 10.1016/j.compstruct.2021.114698, 10.1177/0731684417727143, 10.1002/app.46770, 10.1016/j.porgcoat.2022.107015.

3. Kindly add a table that describes the main physical and chemical properties of the raw materials used in this study.

4. Were the preparation methods described by the authors come in accordance with a certain standard or do they follow previous procedures?

5. Kindly provide the SEM parameters in the characterization section including the accelerating voltage and the working depth.

6. For the shear tests, what was the reason for the specified test conditions in this research? Do the speed of test value relate to a specific application? 

7. How many samples were used per configuration for the shear test? What is the geometry of the sample used for the shear test? A sketch of the sample is required to be added.

8. Was there any relationship between the filler content, flame retardancy and shear strength? Were there any further expectations for the case of using higher weight fractions of the flameretardant?

9. The conclusion needs to be modified to summarize the research outcomes in short statements with clear observations.

Author Response

Response to Reviewer 3 Comments

Dear Editor

We are submitting a revised manuscript entitled “Preparation and characterization of cardanol-based flame retardant for enhancing the flame retardancy of epoxy adhesives”.

We thoroughly read your mail and reviewer’s comments. The kind and meticulous comments of the reviewers were very helpful to our manuscript. In addition, thanks to reviewers sincerely comments, the missing part could be confirmed. Therefore, we included the reviewer’s comments in the manuscript as follows (The revised or added parts in manuscript were written with red letters).

Point 1: #1. In the abstract, add a final statement that highlights the importance of this research and its possible potentials. Also, introduce the problem in the initial lines of the abstract.

Response 1: Following the reviewer’s meticulous comments, final statements in abstract section was added to highlight its potentials in industrial area. Also, the problem of epoxy resin was added in the initial lines of the abstract section. Based on the reviewer’ comments, the abstract section was more clear than before.

; ‘Epoxy resin has a versatile application due to its excellent properties. however, its easily flammable property limits further application.’ was added in the initial lines of the abstract section.

; ‘Based on the findings gathered in … for improving flame retardancy and adhesion property, which will be promising for the industrial area.’

The modified sentences were marked in red in the draft.

Point 2: #2. The introduction needs to be improved by relating to the mechanics of the studied materials and their mechanical characteristics. The references to be included are:   

10.1177/0021998318790093, 10.1016/j.polymertesting.2017.09.009, 10.1016/j.compstruct.2021.114698, 10.1177/0731684417727143, 10.1002/app.46770, 10.1016/j.porgcoat.2022.107015. 

Response 2: Thanks to reviewer’s kind comments, the attached references were carefully confirmed. However, in this study, the introduction section was more focusing on eco-friendly materials than any other subject. The kindly recommendations are highly appreciated, but we do not wish to add introduction on the mechanical characteristics of the studied materials. Also, most of the materials used in the experiments were used as received from Sigma-Aldrich and Daejung chemical.

Point 3: #3. Kindly add a table that describes the main physical and chemical properties of the raw materials used in this study.

Response 3: According to reviewer’s comments, all the physical and chemical properties of the raw materials used in this study was checked again. All the reagents except for the synthesized CDPP were purchased from Sigma-Aldrich or Daejung chemical and used as without further purification. The state information about the synthesized CDPP was added in section 2.2.

; The product obtained as brownish liquid was referred to as cardanyl diphenylphosphate (CDPP).

Point 4: #4. Were the preparation methods described by the authors come in accordance with a certain standard or do they follow previous procedures?

Response 4: Following the reviewer’s meticulous comments, all the preparation methods in this draft was carefully checked again. In the case of LOI measurements, samples follow the ASTM D2863 standard. In the case of cone calorimeter measurements, samples were prepared following the ISO 5660-1 standard. In the case of single lap shear tests, ASTM D1002 standard for single lap shear test was adjusted while maintaining the ratio.

Point 5: #5. Kindly provide the SEM parameters in the characterization section including the accelerating voltage and the working depth.

Response 5: According to the reviewer’s careful comments, the accelerating voltage and working depth was added in the characterization section.

;’Scanning electron microscopy (SEM) images … JEOL JSM-7601F PLUS (accelerating voltage: 15.0 kV, working depth: 8.0 mm).

Point 6: #6. For the shear tests, what was the reason for the specified test conditions in this research? Do the speed of test value relate to a specific application?

Response 6: Thanks to reviewer’s meticulous comments, the specified test conditions for the shear tests was checked again. The shear test conditions, which was applied in this paper was followed ASTM D1002 standard. However, for the lab scale study, it was adjusted while maintaining the ratio. The speed of test value is not related to any particular application. It was adjusted for obtaining the smooth graph.

Point 7: #7. How many samples were used per configuration for the shear test? What is the geometry of the sample used for the shear test? A sketch of the sample is required to be added.

Response 7: Thanks to reviewer’s kindly comments, the reliability of the shear test results was again checked. The sample’s shear test was repeated at least four times. For the shear test, EP series containing CDPP was applied to wood plate. A sketch for dimensions for single lap shear test of EP series was added in manuscript. It was marked as Figure 8.

Point 8: #8. Was there any relationship between the filler content, flame retardancy and shear strength? Were there any further expectations for the case of using higher weight fractions of the flame retardant?

Response 8:

As the filler contents increased, the flame retardancy and shear strength was improved. In the case of the shear strength, the phosphorus group in CDPP induces the improvement of the shear strength, due to the higher polarity of the phosphorus group. In the case of the flame retardancy, the flame retardancy was improved as the contents of CDPP was increased. Based on the results until the contents of CDPP were 5wt%, the material containing phosphorus group will cause the spontaneously improvement of adhesion property and flame retardancy, which will be promising potential in adhesive industrial area. However, when the contents of CDPP is 5wt% or more, CDPP interferes with mechanical interlocking, resulting in deterioration in adhesion property. In the case of EP containing 10wt% or more of CDPP, it is expected that flame retardant performance may be improved. However, in this paper, research was conducted with a focus on identifying the optimal point for excellent adhesive performance and flame retardant performance at the same time.

Point 9: #9. The conclusion needs to be modified to summarize the research outcomes in short statements with clear observations.

Response 9: Thanks to reviewer’s meticulous comments, the conclusion section was revised. The summarized research outcomes were added in the conclusion section, which will be more comprehensible to readers. The modified sentences were marked in red in the draft.

; ‘The pk-HRR value was decreased from 564.07 kW/m2 at 348 s of EP0 to 433.03 kW/m2 at 83 s of EP10. Also, the THR value was decreased from 168.1 MJ/m2 of EP0 to 147.9 MJ/m2 of EP10.’ was added in draft.

; ‘The lap shear strength value was improved from 6.27 MPa of EP0 to 7.18 MPa of EP5,’ was added in draft.

Round 2

Reviewer 1 Report

The authors have addressed my concerns and it can be accepted as it is. 

Reviewer 3 Report

The introduction needs to be improved by relating to the mechanics of the studied materials and their mechanical characteristics. The references to be included are:   

10.1177/0021998318790093, 10.1016/j.polymertesting.2017.09.009, 10.1016/j.compstruct.2021.114698, 10.1177/0731684417727143, 10.1002/app.46770, 10.1016/j.porgcoat.2022.107015.